# The Effect of Carbon Nanotubes on the Strength of Sand Seeped by Colloidal Silica in Triaxial Testing

**DOI:** 10.3390/ma14206119

**Published:** 2021-10-15

**Authors:** Weifeng Jin, Ying Tao, Xin Wang, Zheng Gao

**Affiliations:** 1School of Civil Engineering and Architecture, Zhejiang University of Science and Technology, Hangzhou 310023, China; 212002814015@zust.edu.cn (Y.T.); 211602814002@zust.edu.cn (X.W.); 2Hydro China Huadong Engineering Corporation, Hangzhou 310014, China; gao_z3@ecidi.com

**Keywords:** colloidal-silica-stabilized sand, deviatoric stress at failure, carbon-nanotube-reinforced, triaxial test, shear strength prediction

## Abstract

Colloidal silica can quickly seep through sand and then form silica gels to cement sand particles. To improve the strength of sand seeped by colloidal silica, carbon nanotubes were dispersed in the colloidal silica to form carbon-nanotube-reinforced sand-gel composites. Then triaxial tests were performed to explore how carbon nanotube content affects shear strength. The test results showed that: (1) with the increase of colloidal silica concentration, the shear strength significantly increased with the same carbon nanotube content (especially the low concentration of 10 wt. % colloidal silica, which showed almost no reinforcing effect with carbon nanotubes) while 40 wt. % colloidal silica plus 0.01 wt. % carbon nanotube caused the maximum increase of shear strength by up to 93.65%; (2) there was a concentration threshold of colloidal silica, above which the shear strength first increased to the peak value and then decreased with increasing carbon nanotube content (and we also established a formula to predict such phenomenon); and (3) SEM images showed that carbon nanotubes were connected as several ropes in the micro-cracks of the silica gel, resulting in greater macroscopic shear strength. Our new method of mixing carbon nanotubes and colloidal silica to seep through sand can contribute to sandy ground improvement.

## 1. Introduction

A fast and non-toxic way to stabilize loose sand is the seepage of colloidal silica through sand. The colloidal silica has a low viscosity, which allows its rapid seepage through sand [1,2]. In an alkaline environment, the silica nano-particles in the colloidal silica are stably suspended against gelation due to the repulsive force between the particles. When the pH of colloidal silica is changed from an alkaline to an acid, the repulsive force between the particles decreases, and the silica nano-particles agglomerate to form silica gels which can cement sand particles [3]. Additionally, by adjusting the pH of the colloidal silica, the colloidal silica can maintain its fluidity for a certain period and then become a solid gel [4]. The above characteristics make colloidal silica ideal for stabilizing loose sand by rapid seepage.

Currently, for the effect of adding carbon nanotubes on the sand seeped by colloidal silica, no references are available. For sand seeped by colloidal silica, the existing literature has not provided answers the following two questions: (1) as the carbon nanotube content increases, what are the trends of shear strength at a certain concentration of colloidal silica; and (2) what is the optimal ratio of carbon nanotube to colloidal silica resulting in the maximum shear strength? We describe current knowledge related to our research from three aspects. The first aspect focused on the sand stabilized by colloid silica, especially the seepage and mechanical characteristics of colloidal-silica-stabilized sand. For instance, Gallagherand Lin [1] studied colloidal silica’s ability to transport through sand in an adequate concentration; Fujita and Kobayashi [3] focused on the fundamental transport behaviors of silica particles influenced by pH conditions; Agapoulaki and Papadimitriou [5] studied travel distance and the effect of temperature on viscosity-versus-time curves; Saiers et al. [6] performed tests on transport through heterogeneous porous media; Hamderi and Gallagher [7] studied the simulation of optimum coverage; Hamderi et al. [8] studied the numerical model for simulating colloidal silica transport through sand columns; Hamderi and Gallagher [9] studied the effect of injection rate on the degree of grout penetration; Gallagher et al. [2] used field tests under explosion to study liquefaction mitigation of colloidal-silica-stabilized sand; Gallagher et al. [4] used centrifuge tests to study colloidal-silica-stabilized sand against liquefaction; Pamuk et al. [10] performed centrifuge tests on colloidal-silica-stabilized site; Conlee et al. [11] performed centrifuge modeling for liquefaction mitigation; Conlee [12] performed centrifuge model tests and full-scale field tests on colloidal-silica-stabilized soil; Kodaka et al. [13] modeled strength and cyclic deformation characteristics of colloidal-silica-stabilized sand; Antonio-Izarraras et al. [14] and Díaz-Rodríguez et al. [14] used shear tests to investigate the cyclic strength of sand stabilized with colloidal silica; Kakavand and Dabiri [15] and Wong et al. [16] performed shear tests on sandy soil improved by colloidal silica; Gallagher et al. [17] used triaxial tests to investigate the influence of colloidal silica on cyclic undrained behavior of sand; Mollamahmutoglu and Yilmaz [18] used triaxial tests to studypre- and post-cyclic strength of colloidal-silica-stabilized sand; Persoff et al. [19] investigated the influence of dilution and contaminants on the strength of colloidal-silica-stabilized sand; Pavlopoulou et al. [20] used monotonic and cyclic loading tests to compare the differences between colloidal-silica-stabilized sand and untreated sand in the stress-strain relationships and cumulative strain curves; Triantafyllos et al. [21] used triaxial compression tests to find important changes in the colloidal-silica-stabilized-sand’s mechanical behavior such as increase in stress ratio and relocation of sand’s critical state line in the e-p’ plane; Krishnan et al. [22] obtained the gel time by adding salt and varying pHs and used direct shear tests and triaxial tests to analyze the mechanical enhancement of colloidal-silica-stabilized sand; Vranna et al. [23] used undrained monotonic and cyclic triaxial tests to obtain monotonic and cyclic strength of colloidal-silica-stabilized sand; and Ghadr et al. [24] used undrained triaxial tests to determine the shear strength and the critical-state-line changes of colloidal-silica-stabilized sand. However, the above researchers did not extended their research to the application of carbon nanotubes in colloidal-silica-stabilized sand, so this literature cannot answer how carbon nanotubes affect the strength of colloidal-silica-stabilized sand. The second aspect of the state of the art related to our research is the silica gel containing carbon nanotubes. Gavalas et al. [25] described a class of composite materials designed by combining multiwall carbon nanotubes and silica gel. However, this gel has not been used to stabilize sand. The third aspect of the state of the art related to our research is the application of carbon nanotubes in the Portland cement matrix. In previous literatures, carbon nanotubes have been dispersed in Portland cement and used to bridge micro-cracks in the matrix. For instance, Wang et al. [26] studied the durability of cement doped with carbon nanotubes; Liu et al. [27] studied the shrinkage and crack resistance of carbon-nanotube-reinforced cement; Cwirzen et al. [28] investigated the mechanical properties of cement reinforced by surface-decorated carbon nanotubes; Li et al. [29] studied the pressure-sensitive properties and microstructures of carbon-nanotube-reinforced cement; Nochaiya and Chaipanich [30] investigated the effect of carbon nanotubes on the porosity and microstructure of cement; Rana and Fangueiro [31] review the dispersion and mechanical properties of carbon-nanotube-reinforced cement; and Li et al. [32] studied the coupling effect of nano-silica sol-gel and carbon nanotubes on the cement based materials. However, the previous literature mixed carbon-nanotube-dispersed Portland cement and sand by stirring, and such cement is not suitable to seep through sand due to its high viscosity. So, the main difference from these previous literatures is that we transport carbon nanotubes to the pores between sand particles by seepage instead of stirring. In conclusion, the research gap between the previous literature and our work is that no references elaborated the influence of carbon nanotube on the strength of sand seeped by colloidal silica, since previous studies did not extend the application of carbon nanotubes to the colloidal-silica-stabilized sand. Besides, the case of carbon-nanotube-dispersed Portland cement cannot be a direct reference in our case, since the mixing method of Portland cement and sand was stirring, and not the seepage as in our case.

The contribution of this paper is that, for sand seeped by colloidal silica, we obtained laws concerning how carbon nanotube content affects the strength of colloidal-silica-stabilized sand. That is, we obtained the trends of strength with increasing carbon nanotube content at different concentrations of colloidal silica. We also obtained the optimal ratio of carbon nanotube to colloidal silica resulting in the maximum shear strength. Additionally, we established a formula to predict strength varying with carbon nanotube content at 40 wt. % colloidal silica. The motivation of this paper is that, based on the images of the scanning electron microscope (SEM), there are a large number of micro-cracks in the silica gel which bonds sand particles. These micro-cracks reduce the strength of the gel-sand composite. So, we attempted to add carbon nanotubes to colloidal silica in the hope that carbon nanotubes in the micro cracks could result in the improvement of the macro shear strength of the gel-sand composite. The reason for doing this research is that the effects of carbon nanotube content on the shear strength can be clarified through tests. In addition, what this article would add to the current available literature are the laws of how carbon nanotube content affects colloidal-silica-stabilized sand, as well as the prediction formula at high concentration of colloidal silica (thisis the first time that the characteristics of the trends of shear strength with increasing carbon nanotubes are clarified). So, this paper is suitable for the application in the field of ground improvement by using colloidal silica, since we present the optimal ratio of colloidal silica to carbon nanotubes resulting in the maximum shear strength as well as a prediction formula for the effect of carbon nanotubes on the strength of colloidal-silica-stabilized sand.

In this paper, based on the triaxial tests, we first explore the influence of carbon nanotube content on the shear strength of sand seeped with different concentration levels of colloidal silica; second, we introduce physically meaningful variables to propose a formula for predicting the maximum shear strength; finally, based on the SEM photos, from the micro scale perspective, we explore the morphology of carbon nanotubes in the micro-cracks of silica gel, which results in greater macro shear strength compared to sand stabilized without carbon nanotubes.

## 2. Experimental Details

Gallagher et al. [2] described various factors influencing the gel time of colloidal silica such as concentration, nano-silica particle size, ionic strength and pH. Gel time decreases with increasing concentration, increasing the size of the nano-silica particle, and increasing ionic strength, and gel time reaches its minimum when pH ranges from 5 to 7. In our tests, we adjusted the pH value to control the gel time, since the particle size was not changed and we did not add salt to control ionic strength. For the four concentrations of colloidal silica used in our tests, when the pH is adjusted to 5~6, colloidal silica forms solids and cement sand particles within one day. So, we adjusted the pH value to 5~6 before using colloidal silica to seep through sand specimens. After the sand specimens were seeped by colloidal silica, the sand specimens were cured for three days to ensure that the specimens were stabilized. Then we performed triaxial tests on these specimens.

### 2.1. Experimental Materials

The sand used was Pingtan standard sand from Fujian, China. Index properties of the sand, including relative density, are listed in Table 1.

The multi-walled carbon nanotubes (MWCNTs), produced by Suzhou Hengqiu Graphene Technology Co., Ltd., (Suzhou, China) were used as reinforcing fibers. The physical and structural properties of MWCNTs are listed in Table 2. The MWCNTs have inner diameters of 3–5 nm, outer diameters of 8–15 num, and lengths of 3–12 ìm. The specific surface area is >230 m^2^/g according to the manufacturer.

The colloidal silica used was provided by Qingdao Maike Silica Gel Dessicant Co., Ltd., and four concentrations of 10, 20, 30, and 40% by weight were used. The physical properties of the colloidal silica are shown in Table 3. The silicanano-particles, which are suspended under an alkaline environment, have diameters in the range of 10 to 20 nm.

### 2.2. Specimen Preparation and Testing Apparatus

For the method of specimen preparation, we used the carbon-nanotube-dispersed colloidal-silica to seep through and then stabilize the sand. The key point, which was to disperse carbon nanotubes in colloidal-silica, was fulfilled within two dispersion steps: first, we put carbon nanotubes in the container filled with colloidal-silica, and then used a rotating bar to mechanically stir the colloidal-silica and carbon nanotubes. Secondly, ultrasonic vibration conducted across the container and make carbon nanotubes more uniformly dispersed in colloidal-silica. The duration time of each of the above dispersion step was calibrated by trial. We used the above two steps to make sure that carbon nanotubes were well dispersed, which was why we used the above two steps to disperse carbon nanotubes.

For the method of testing shear strength, since triaxial testing apparatuses can maintain a user-defined stress in the horizontal direction and apply load in the vertical direction, we utilized triaxial apparatuses to obtain the shear strength with different horizontal stresses.

The sand specimens were treated using the following steps. First, the pH of colloidal silica was adjusted to 5.0–5.5 by adding acetic acid, then the colloidal silica was magnetically stirred with carbon nanotubes for 30 min (see Figure 1a). Secondly, carbon nanotubes were further dispersed by ultrasonic dispersion for 60–120min (see Figure 1b). Finally, by a peristaltic pump, carbon-nanotube-dispersed colloidal-silica was slowly injected into the sand from the bottom of a cylindrical mold (see Figure 1c). The stabilized specimens are shown in Figure 2. The reason for moisture proof membranes is to prevent the moisture in the specimen from evaporating. Although it is not necessary to immerse the specimen in water, we still ensure that the specimen is in a wet state and moisture will not leak from the membrane joints, so we still placed the specimen wrapped with the membrane in water.

AGDS advanced triaxial system (see Figure 3) was used to carry out undrained triaxial compression tests on specimens. The diameter and height of the test specimen are 38 mm and 76 mm, respectively.

### 2.3. Experimental Plan

The experimental plan, as summarized in Table 4, is mainly aimedat investigating the effect of the content of carbon nanotube on the strength of the stabilized specimens. Sands were seeped and stabilizedby the mixture of colloidal silica and carbon nanotubes. The concentrations of colloidal silica are 10, 20, 30 and 40 wt. %, respectively. The contents of carbon nanotubes are 0, 0.01, 0.02, 0.03, 0.04 and 0.05% by weight of colloidal silica, respectively. The initial mean effective stresses in undrained triaxial tests for these specimens were 50, 80 and 110 kPa, respectively. During each triaxial test, the confining pressure (horizontal stress) is set equal to the initial mean effective stress. The molds containing the stabilized specimens were wrapped in moisture proof membranes and immersed in water to cure for three days before testing. Thus, there are a total of 72 specimens tested.

## 3. Testing Results and Analysis

Figure 4 shows the failure modes of the specimens after the triaxial tests. The specimens in Figure 4a–c were reinforced with 10, 20 and 30 wt. % colloidal silica respectively, while the sample in Figure 4d was reinforced with 40 wt. % colloidal silica mixed with 0.02% carbon nanotubes. It can be seen from Figure 4a–c that as the concentration of colloidal silica increases, the slip surface becomes more obvious. In Figure 4d, the stabilized specimen with carbon nanotubes shows the most obvious slip surface.

### 3.1. Effect of Carbon Nanotubes on the Shear Strength

Since we use deviatoric stress at failure as shear strength, Figure 5a–c show the curves of shear strength versus carbon nanotube content under different initial mean effective stresses of 50, 80 and 110 kPa. The corresponding data are shown in Table 5. As shown in Figure 5, when the concentration of colloidal silica is 10 wt. %, carbon nanotubes have no reinforcement effect on the specimens. For 20 wt. % colloidal silica, carbon nanotubes cause a slight increase in the deviatoric stress at failure, which is less than the increase in 30 wt. % colloidal silica. In the case of 40 wt. % colloidal silica, 0.01–0.02 wt. % carbon nanotubes cause the greatest reinforcement effect on the specimens (i.e., under the initial mean effective stresses of 50, 80 and 110 kPa, the deviatoric stresses at failure increase by 83.6, 93.7 and 78.2%, respectively). For 0.01 wt. % carbon nanotubes dispersed in 10, 20, 30 and 40 wt. % colloidal silica, according to Table 5 or Figure 5, the shear strength increased by up to 9.93, 13.1, 21.37 and 93.65%, respectively. That is, as the concentration of colloidal silica increases from 10 to 40 wt. %, after adding 0.01 wt. % carbon nanotubes, and the maximum percentage of shear strength increment increases from 9.93 to 93.65%. Therefore, the reinforcement effect of carbon nanotubes increases with increasing colloidal silica concentration. Additionally, for 40% colloidal silica, based on different horizontal stress levels, there are optimal contents of carbon nanotubes between 0.01–0.02 wt. % which lead to the peak shear strengths.

Since the above-mentioned optimal carbon nanotube contents only appear in 40 wt. % colloidal silica, and does not appear in 10, 20 and 30 wt. % colloidal silica, we can conclude that there is a threshold for the concentration of colloidal silica, above which there exists the phenomenon of the optimal carbon nanotube content leading to the peak of shear strength. Obviously, 40 wt. % is above this threshold of colloidal silica concentration.

Previous literature has shown that, for Portland-cement-based matrix, with the increase of carbon nanotube content there are three possible trends in the compressive strength: (1) the compressive strength remains unchanged or even decreases [33,34]; (2) the compressive strength increases [35,36,37,38,39,40,41,42]; or (3) the compressive strength increases first and then decreases [43,44,45]. Our study shows similar trends in carbon-nanotube-reinforced sand-gel composites, and such trends depend on the concentrations of colloidal silica, as shown in Figure 5: (1) at 10 wt. % colloidal silica concentration, the shear strength even decreases with increasing carbon nanotube content; (2) when the colloidal silica concentration is 20 wt. % or 30 wt. %, the increase of carbon nanotube content causes the shear strength to increase; and (3) at 40 wt. % colloidal silica concentration, with the increase of carbon nanotube content, the shear strength first increases and then decreases (i.e., for a given horizontal stress, there is an optimal carbon nanotube content that leads to the peak shear strength during vertical compression).

Figure 6 shows the curves of the internal friction angle versus carbon nanotube content, and the corresponding data are shown in Table 6. Figure 7 shows the curves of cohesion versus carbon nanotube content, and the corresponding data are shown in Table 7. If the data point of 0.05% carbon nanotubes in 40 wt. % colloidal silica is not considered, then it can be concluded that as the carbon nanotube content increases, the internal friction angle first decreases and then increases, which reaches its minimum at 0.03% carbon nanotube content (see Figure 6). On the contrary, at 0.03% carbon nanotube content, cohesion reaches its maximum (see Figure 7). That is, as the content of carbon nanotubes increases, and the internal friction angle and cohesion have opposite development trends.

In the next section, for the case where there is an optimal carbon nanotube content in 40 wt. % colloidal silica, a formula reflecting the peak of shear strengthwith the optimal carbon nanotube content is established.

### 3.2. Shear Strength Model Reflecting Optimal Carbon Nanotube Content

#### 3.2.1. Model Establishment

Here we present a formula for the effect of the optimal content of carbon nanotubes on the strength of sands stabilized with the mixture of colloidal silica and carbon nanotubes. Since the optimal carbon nanotube content which leads tothe peak shear strength is only observed in the case of 40 wt. % colloidal silica, while this effect of the optimal carbon nanotube content is not observed at lower colloidal silica concentrations (see Figure 5), we establish the formula only for sands stabilized by colloidal silica at the concentration of 40 wt. %.

Suppose that the shear strength of cemented sand is composed of two parts, the basic part is caused by the combination of colloidal silica and sand particles, and the remaining part is caused by carbon nanotubes.
(1)$$q_{f}=q_{f/CS}+q_{f/CNTs}$$
where $q_{f}$ is the deviatoric stress at failure of cemented sand, $q_{f/CS}$ is the part caused by the combination of colloidal silica and sand, and $q_{f/CNTs}$ is the part caused by carbon nanotubes.

Next, we give the expressions of $q_{f/CS}$ and $q_{f/CNTs}$.

First, we express $q_{f/CS}$ as a function of friction angle and colloidal silica concentration, which is inspired by the expression of cemented sand [46].
(2)$$q_{f/CS}=\frac{2\sin\varphi }{1-\sin\varphi }{p^{\prime }}_{initial}+k_{1}C_{CS}$$
where φ is the internal frictional angle of sands treated with 40 wt. % colloidal silica plus 0% carbon nanotubes, ${p^{\prime }}_{inital}$ is the initial mean effective stress, $k_{1}$ is the parameter relating the colloidal silica concentration to the deviatoric stress at failure, and $C_{CS}$ is the colloidal silica concentration.

Then we provide a function of $q_{f/CNTs}$ to express the effect of carbon nanotubes on the shear strength by introducing two key parameters. The first key parameter is the optimal carbon nanotube content, which is denoted as $C_{CNTs}^{o}$ since, as shown in Figure 5, for a given initial mean effective stress, there is an optimal carbon nanotube content to maximize the shear strength (deviatoric stress at failure)at 40 wt. % colloidal silica concentration. The second key parameter is the increment of deviatoric stress at failure with the optimal carbon nanotube content, which is denoted by $q_{\Delta }$ and expressed as follows
(3)$$q_{\Delta }=q_{C_{CNTs}^{}=C_{CNTs}^{o}}-q_{C_{CNTs}^{}=0\%}$$
(4)$$q_{f/CNTs}=\frac{2q_{\Delta }\cdot C_{CNTs}^{o}\cdot C_{CNTs}}{{\left(C_{CNTs}\right)}^{2}+{\left(C_{CNTs}^{o}\right)}^{2}}$$
where $q_{C_{CNTs}^{}=C_{CNTs}^{o}}$ is the deviatoric stress at failure with the optimal carbon nanotube content, while $q_{C_{CNTs}^{}=0\%}$ is the deviatoric stress at failure without carbon nanotubes. By introducing the above two key parameters, $q_{f/CNTs}$, which reflects the effect of carbon nanotubes on the shear strength, is expressed as follows

This is the first time that Equation (4) is designed in order to fulfill the requirement of describing the phenomenon that the shear strength first increases and then decreased with increasing carbon nanotube content in out tests. 

Here is how we arrived at the idea to present Equation (4). As shown in Figure 8, with the increase of carbon nanotube content, the shear strength increases first and then decreases according to the test data. Therefore, in order to show the above characteristics, when we tried to express $q_{f/CNTs}$ (part of shear strength caused by carbon nanotubes) as a function of $C_{CNTs}^{}$ (carbon nanotube content), we found that the reciprocal of hyperbolic function can describe the characteristics of the peak shear strength. So we designed Equation (4) as the form of the reciprocal of hyperbolic function. Then Equation (4) can simultaneously describe the optimal carbon nanotube content and its corresponding peak shear strength.

#### 3.2.2. Determination of Parameters

φ is the internal frictional angle of sand treated with 40 wt. % colloidal silica plus 0% carbon nanotubes, and the value of φ can be obtained from the triaxial compression tests.

In order to obtain the value of $k_{1}$, Equation (2) is rewritten as Equation (5) to calculate $k_{1}$. The variables on the right side of Equation (5) are all obtained from tests on specimens stabilized only with colloidal silica. Utilizing Equation (5), $k_{1}$ is obtained as the average value of tests with different initial mean effective stresses.
(5)$$k_{1}=\frac{1}{C_{CS}}\left(q_{f/CS}-\frac{2\sin\varphi }{1-\sin\varphi }{p^{\prime }}_{initial}\right)$$

$C_{CNTs}^{o}$ is the optimal carbon nanotube content when the deviatoric stress at failure reaches its maximum value. As shown in Figure 5, in the case of 40 wt. % colloidal silica, for the initial mean effective stresses of 50, 80 and 110 kPa, the deviatoric stresses at failure reaches their maximum values at the carbon nanotube contents of 0.02, 0.01 and 0.01 wt. %, respectively. We use Equation (6) to obtain $C_{CNTs}^{o}$ in the prediction.
(6)$$C_{CNTs}^{o}=\frac{m_{1}}{{\left({p^{\prime }}_{0}\right)}^{m_{3}}}+m_{2}$$
where $m_{1}$, $m_{2}$, and $m_{3}$ are parameters. By curve fitting as shown in Figure 9, $m_{1}$, $m_{2}$ and $m_{3}$ are determined as 1.95 × 10^11^, 9.99 × 10^−5^ and 9, respectively.

$q_{\Delta }$ is the increment of deviatoric stress at failure with the optimal carbon nanotube content. In specimens stabilized with 40 wt. % colloidal silica, for the same initial mean effective stress, substituting the values of $q_{C_{CNTs}^{}=C_{CNTs}^{o}}$ and $q_{C_{CNTs}^{}=0\%}$ from Figure 5 into Equation (3) yields the value of $q_{\Delta }$. There are three values of $q_{\Delta }$ for the three different initial mean effective stresses ${p^{\prime }}_{initial}$. We express $q_{\Delta }$ as a polynomial function of ${p^{\prime }}_{initial}$, as shown in Equation (7), and Figure 10 shows that this polynomial form fits the test data well.
(7)$$q_{\Delta }=a\times {\left({p^{\prime }}_{initial}\right)}^{2}+b\times {p^{\prime }}_{initial}+c$$
where a, b, and c are parameters. By curve fitting, a, b, and c are determined as −0.057, 12.07, and 16.77, respectively.

Parameters used for the prediction of variation of shear strength with the content of carbon nanotubes are listed in Table 8.

#### 3.2.3. Prediction of Shear Strength

The prediction of variations of deviatoric stresses at failure with the content of carbon nanotubes is in good agreement with the tests, as shown in Figure 11, reflecting the main characteristic of the effect of carbon nanotube content on the shear strength: there exists an optimal carbon nanotube content which leads to the peak shear strength. That is, by introducing two physically meaningful parameters (namely the optimal carbon nanotube content and the increment of deviatoric stress at failure with the optimal carbon nanotube content), our model depicts the phenomenon well: for the concentration of 40% colloidal silica, as the carbon nanotube content increases, the shear strength first increases and then decreases.

## 4. Microscale Analysis

Figure 12a,b shows the scanning electron microscope images of the stabilized sand with and without carbon nanotubes. Obviously, there are cracks in the silica gel around the sand particles. However, the cracks in the specimen stabilized by the mixture of colloidal silica and carbon nanotubes are not obviously less than that in the specimen without carbon nanotubes, which cannot definitely indicate that the addition of carbon nanotubes resulted in fewer cracks in the silica gel. Due to the uneven height caused by sand particles, it is difficult to directly focus on the cracks after continuing to zoom in and take pictures. So, it is difficult to observe the carbon nanotubes in the cracks at this time.

As a comparison, silica gel was formed by 40% colloidal silica without carbon nanotubes, whose cracks are shown in Figure 13. In order to further observe the carbon nanotubes in the silica gel cracks, silica gel was formed by dispersing 0.02% carbon nanotubes into 40% silica colloidal without sand, whose cracks are shown in Figure 14.

The phenomenon that carbon nanotubes act as bridges across cracks and inhibit cracking can be seen in Portland-cement-based matrix [29,31]. Similarly, Figure 14 shows that in the gel matrix, the carbon nanotubes are entangled into ropestobridge the crack. From the perspective of the micro-scale, the addition of carbon nanotubes results in ropes entangled by carbon nanotubes in the micro-cracks, and the corresponding macro phenomenon is that shear strength increases with the addition of carbon nanotubes.

## 5. Conclusions

The study of sands seeped and stabilized with the mixture of colloidal silica and carbon nanotubes has allowed the authors to establish the following conclusions:The degree of reinforcement of carbon nanotubes depends on the concentration of colloidal silica; when the same content of carbon nanotubes is added, shear strength increases with increasing concentration of colloidal silica. The low concentration (10 wt. %) of colloidal silica especially shows almost no reinforcement effect after adding carbon nanotubes.However, when the concentration of colloidal silica is increased to 40 wt. %, the shear strength, which can be represented by the deviator stress at failure, can be increased by up to 93.65% after adding 0.01 wt. % carbon nanotubes.There is an optimal carbon nanotube contentleading to the peak shear strength, which only occurs when the colloidal silica concentration is 40 wt. %. That is, there is a concentration threshold of colloidal silica, above which the shear strength increases first and then decreases with increasing content of carbon nanotubes.A formula is established to predict the peak effect of shear strength wellin 40 wt. % colloidal silica by introducing two physically meaningful parameters, which are the optimal carbon nanotube content and the increment of deviatoric stress at failure with the optimal carbon nanotube content.SEM photos show that ropes formed by entangled carbon nanotubes result in enhancing macro strength.The present work is helpful to the design and engineering application of sandy ground stabilization. That is, for the project to improve the strength of the sandy ground, based on the stress level, we present the optimal combination of 40 wt. % colloidal silica and 0.01–0.02 wt. % carbon nanotubes. Besides, the formula of shear strength that varies with carbon nanotube content can be applied to site reinforcement design.

In this paper, there are at least two limitations to the trends of the effects of carbon nanotube content on colloidal-silica-stabilized sand: (1) the carbon nanotubes used here are multi-walled carbon nanotubes, while we do not know how single-walled carbon nanotubes would affect the shear strength of colloidal-silica-stabilized sand; and (2) here we only use magnetically stirring and ultrasonic vibration to achieve the dispersion of carbon nanotubes in colloidal silica. However, different dispersion methods, such as treating carbon nanotubes with acid or using non-covalent surface modification for carbon nanotube surfaces [41], may lead to different trends of shear strength varying with increasing carbon nanotubes.

The guiding research question is how carbon nanotubes affect the shear strength of colloidal-silica-stabilized sand. For this research question, we performed triaxial tests on these carbon-nanotube-reinforced sand-gel composites and obtained the shear strength varying with different concentrations of colloidal silica combined with different carbon nanotube contents. In addition, the optimal ratio of colloidal silica to carbon nanotubes was obtained and a corresponding prediction formula was presented. 

For future research, we can try single-walled carbon nanotubes and different methods of dispersing carbon nanotubes in the colloidal silica to obtain how dispersion methods affect the trends of the shear strength of the carbon-nanotube-reinforced colloidal-silica-stabilized sand. 

## Figures and Tables

**Figure 1 materials-14-06119-f001:**
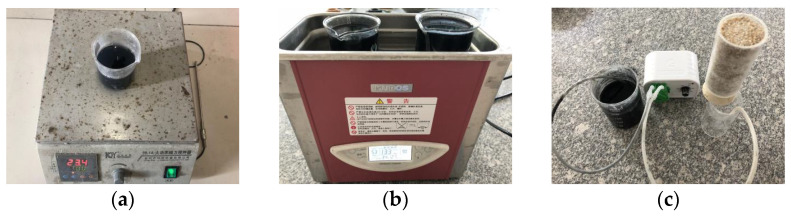
Specimen preparation: (**a**) Magnetically stirred mixture of carbon nanotubes and colloidal silica, (**b**) ultrasonic dispersion of carbonnanotubes in colloidalsilica, (**c**) thecarbon-nanotube-dispersed colloidal-silica seeped through sand.

**Figure 2 materials-14-06119-f002:**
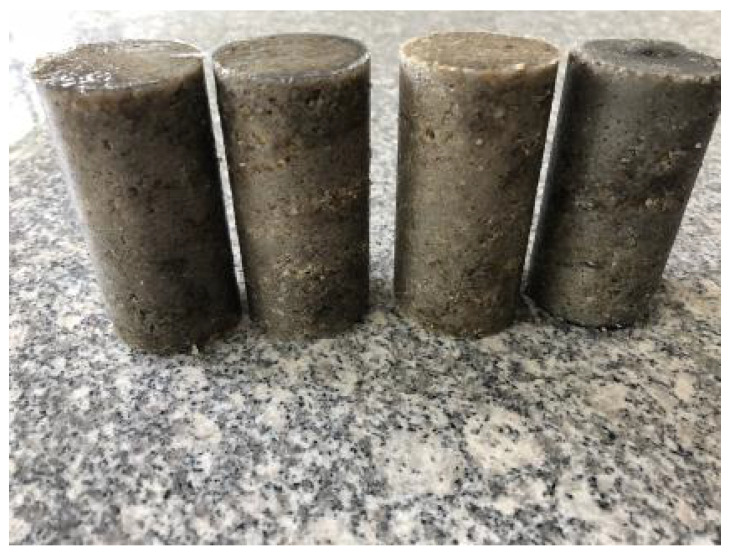
Stabilized specimens.

**Figure 3 materials-14-06119-f003:**
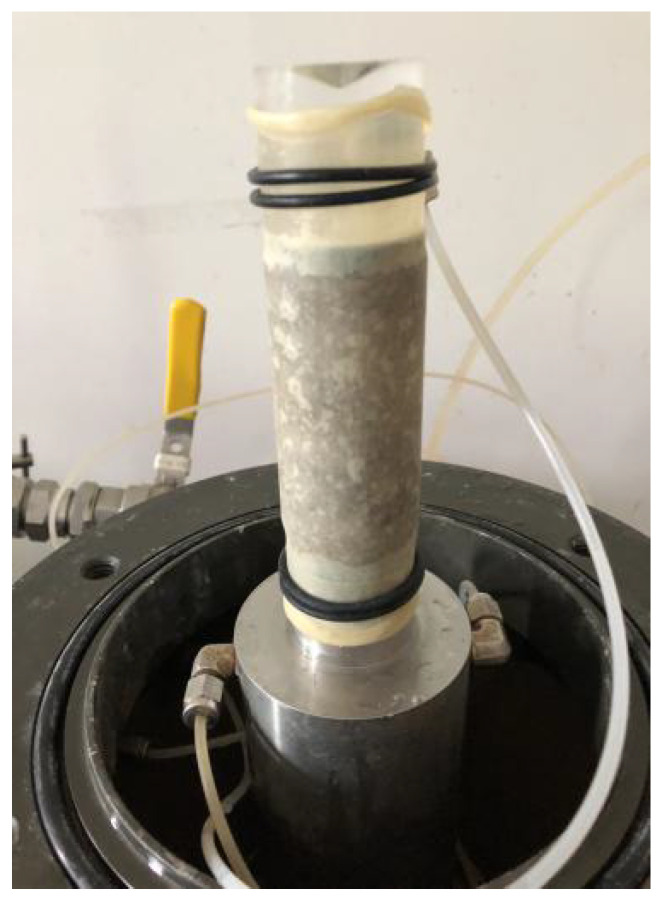
Specimen for triaxial compression test.

**Figure 4 materials-14-06119-f004:**
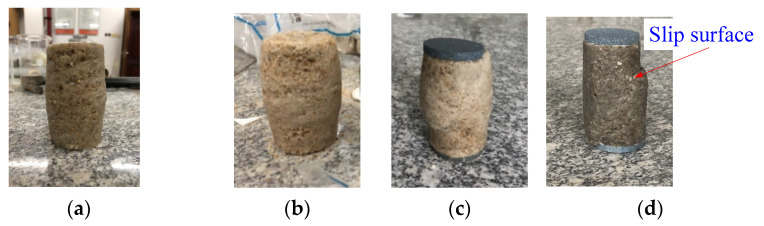
Failure modes of specimens after triaxial tests, and (**a**) 10 wt. % colloidal silica, (**b**) 20 wt. % colloidal silica, (**c**) 30 wt. % colloidal silica, and (**d**) 40 wt. % colloidal silica +0.02 wt. % carbon nanotubes.

**Figure 5 materials-14-06119-f005:**
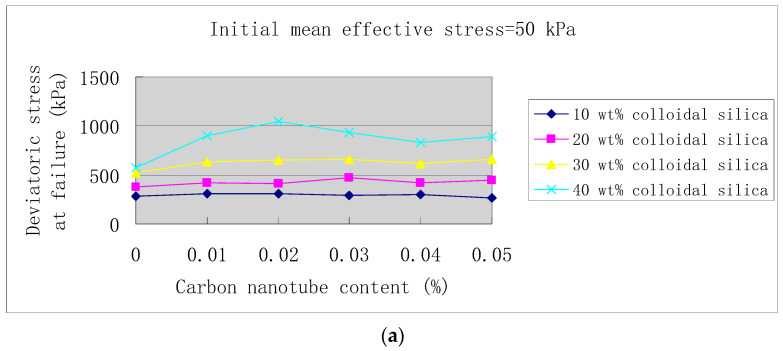
Deviatoric stress at failure vs. carbon nanotube content, and (**a**) initial mean effective stress= 50 kPa, (**b**) initial mean effective stress = 80 kPa, and (**c**) initial mean effective stress = 110 kPa.

**Figure 6 materials-14-06119-f006:**
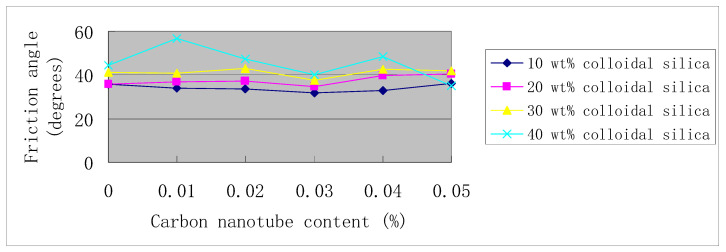
Friction angle vs. carbon nanotube content.

**Figure 7 materials-14-06119-f007:**
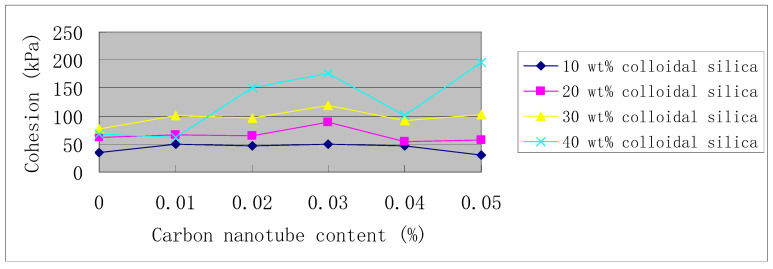
Cohesion vs. carbon nanotube content.

**Figure 8 materials-14-06119-f008:**
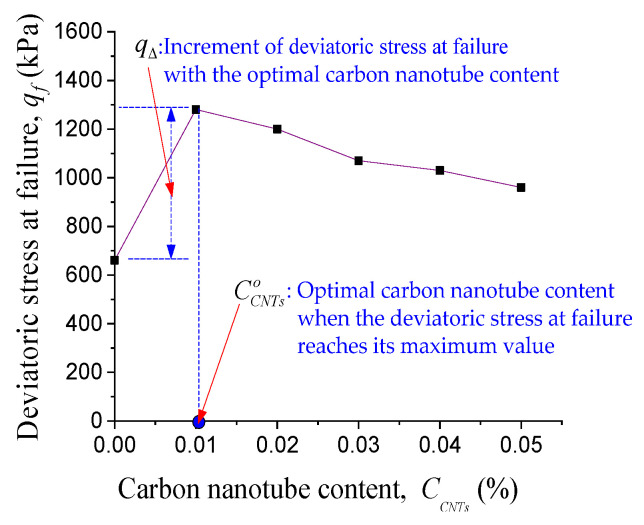
Schematic view of defining two key parameters in $q_{f/CNTs}$ (Equation (4)).

**Figure 9 materials-14-06119-f009:**
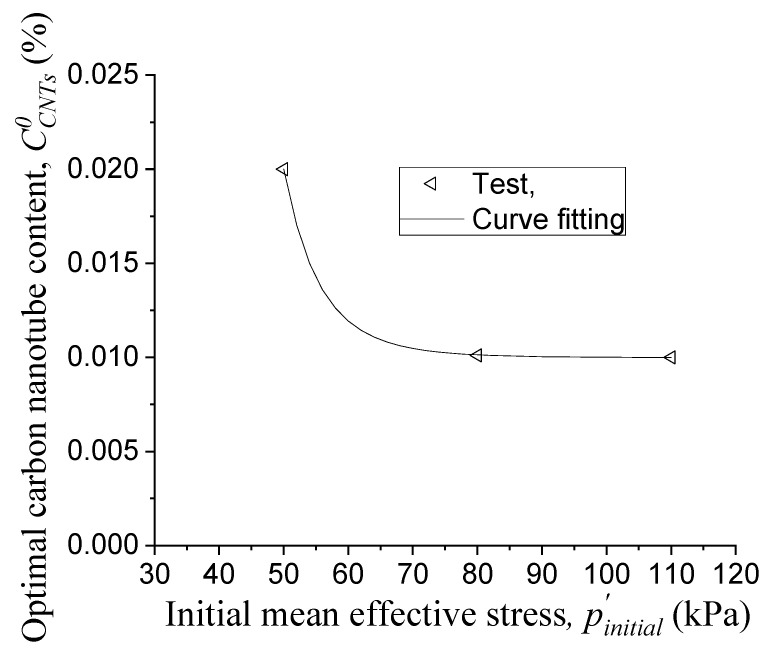
Optimal carbon nanotube content vs. initial mean effective stress.

**Figure 10 materials-14-06119-f010:**
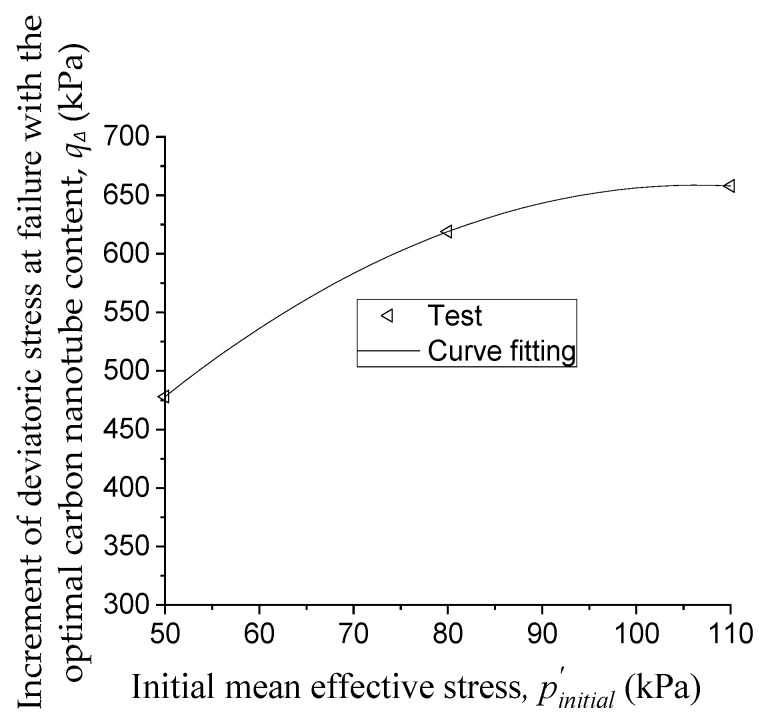
Increment of shear strength with the optimal carbon nanotube content vs. initial mean effective stress.

**Figure 11 materials-14-06119-f011:**
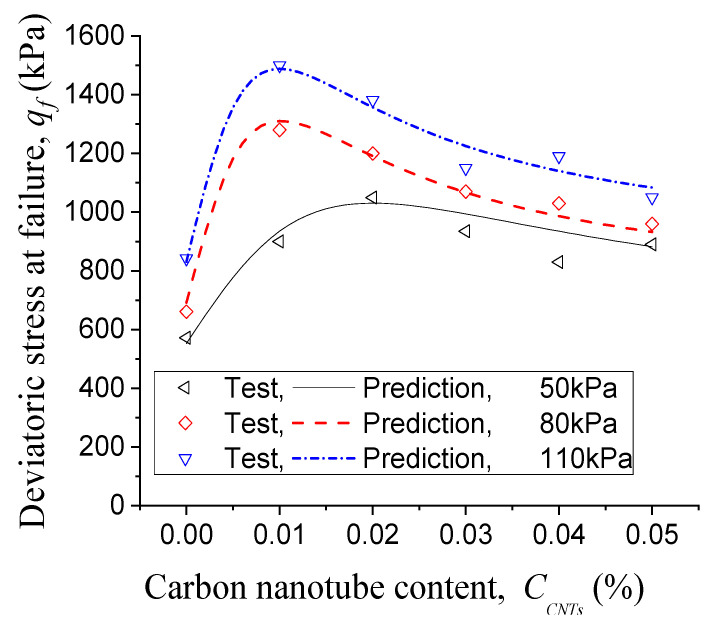
Effects of carbon nanotube content on the shear strength (sands seeped with 40 wt. % colloidal silica).

**Figure 12 materials-14-06119-f012:**
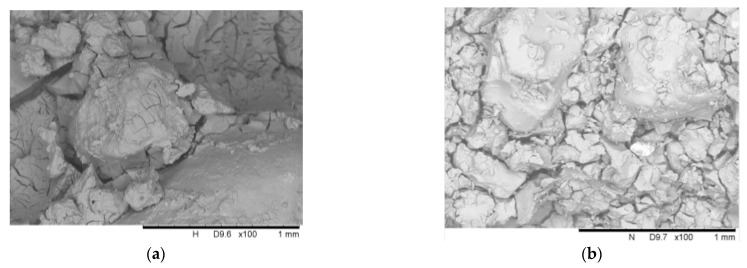
SEM photographs (100× enlargement) of stabilized sand with and without carbon nanotubes: (**a**) sand stabilized by 40 wt. % colloidal silica and (**b**) sand stabilized by 40 wt. % colloidal silica + 0.02% carbon nanotubes.

**Figure 13 materials-14-06119-f013:**
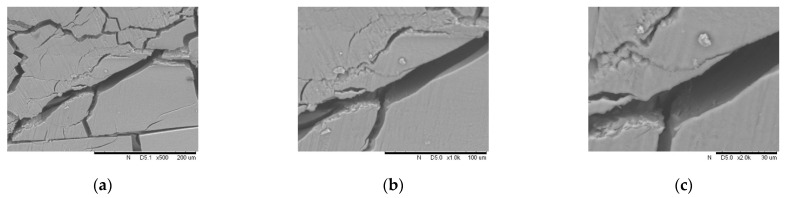
Cracks in silica gel (formed by 40 wt. % colloidal silica): (**a**) 500× enlargement, (**b**) 1000× enlargement, and (**c**) 2000× enlargement.

**Figure 14 materials-14-06119-f014:**
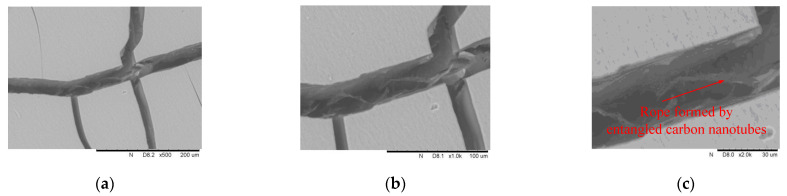
Cracks in silica gel (formed by 40 wt. % colloidal silica + 0.02% carbon nanotubes): (**a**) 500× enlargement, (**b**) 1000× enlargement, and (**c**) 2000× enlargement.

**Table 1 materials-14-06119-t001:** Properties of sand sample.

Specific Gravity, G_s_	Max. Dry Density (kg/m^3^)	Min. Dry Density (kg/m^3^)	Relative Density	Coefficient of Uniformity, C_u_	Coefficient of Curvature, C_c_
2.65	1750	1510	0.43	4	0.51

**Table 2 materials-14-06119-t002:** Properties of multi-walled carbon nanotubes.

Inner Diameters (nm)	Outer Diameter (nm)	Length (μm)	Specific Surface Area (m^2^/g)	Density (g/cm^3^)
3–5	8–15	3–12	>230	2.1

**Table 3 materials-14-06119-t003:** Physical properties of the colloidal silica.

SiO_2_ (%)	pH	Density (g/cm^3^)	Viscosity (Pa·s)	Average Particle Size (nm)
10%	8.5~9.5	1.08~1.10	3.0 × 10^−3^	10~20
20%	8.5~10.0	1.12~1.14	5.0 × 10^−3^	10~20
30%	8.5~10.0	1.19~1.21	7.0 × 10^−3^	10~20
40%	9.0	1.28~1.3	25.0 × 10^−3^	10~20

**Table 4 materials-14-06119-t004:** Experimental plan.

Colloidal Silica Concentration (wt. %)	Carbon Nanotube Content (wt. %)	Initial Mean Effective Stress (kPa)	Curing Period (Days)
10, 20, 30 and 40	0, 0.01, 0.02, 0.03, 0.04 and 0.05	50, 80, and 110	3

**Table 5 materials-14-06119-t005:** Deviatoric stress at failure.

Colloidal Silica Concentration	Curing Period (Days)	Initial Mean Effective Stress (kPa)	Deviatoric Stress at Failure (kPa)					
			Carbon Nanotube 0 wt. %	Carbon Nanotube 0.01 wt. %	Carbon Nanotube 0.02 wt. %	Carbon Nanotube 0.03 wt. %	Carbon Nanotube 0.04 wt. %	Carbon Nanotube 0.05 wt. %
10 wt. %	3	50	282	310	305	292	302	262
10 wt. %	3	80	360	385	368	355	340	358
10 wt. %	3	110	452	461	454	425	440	434
20 wt. %	3	50	374	423	408	472	417	448
20 wt. %	3	80	497	492	523	555	510	535
20 wt. %	3	110	535	601	591	630	630	670
30 wt. %	3	50	524	636	650	656	620	660
30 wt. %	3	80	660	725	800	716	780	803
30 wt. %	3	110	752	860	905	840	870	902
40 wt. %	3	50	572	900	1050	935	830	890
40 wt. %	3	80	661	1280	1200	1070	1030	960
40 wt. %	3	110	842	1500	1382	1150	1190	1050

**Table 6 materials-14-06119-t006:** Internal friction angle.

Colloidal Silica Concentration	Curing Period (Days)	Internal Friction Angle (Degrees)					
		Carbon Nanotube 0 wt. %	Carbon Nanotube 0.01 wt. %	Carbon Nanotube 0.02 wt. %	Carbon Nanotube 0.03 wt. %	Carbon Nanotube 0.04 wt. %	Carbon Nanotube 0.05 wt. %
10 wt. %	3	35.9	33.9	33.7	31.8	32.9	36.1
20 wt. %	3	35.8	36.8	37.4	34.8	39.8	40.6
30 wt. %	3	41.1	40.8	43	37.7	42.8	42.1
40 wt. %	3	44.3	56.7	47.3	40.2	48.6	34.9

**Table 7 materials-14-06119-t007:** Cohesion.

Colloidal Silica Concentration	Curing Period (Days)	Cohesion (kPa)					
		Carbon Nanotube 0 wt. %	Carbon Nanotube 0.01 wt. %	Carbon Nanotube 0.02 wt. %	Carbon Nanotube 0.03 wt. %	Carbon Nanotube 0.04 wt. %	Carbon Nanotube 0.05 wt. %
10 wt. %	3	35.1	49.1	47.1	50.1	46.2	30.8
20 wt. %	3	62.2	66.4	64.3	88.9	54.8	57.9
30 wt. %	3	77.1	100.5	96.3	119.4	91.2	103
40 wt. %	3	67.9	61.4	149.9	176.8	101	196.3

**Table 8 materials-14-06119-t008:** Parameters for predicting the shear strength.

Internal Friction Angle, φ (Degrees)	$k_{1}$ (kPa)	Optimal Carbon Nanotube Content, $C_{CNTs}^{o}$ (wt. %)			Increment of Deviatoric Stress at Failure, $q_{\Delta }$ (kPa)		
		$m_{1}$	$m_{2}$	$m_{3}$	a	b	c
44.3	802.8	1.95 × 10^11^	9.99 × 10^−5^	9	−0.057	12.07	16.77

## Data Availability

All the test data for strength curves in Figure 5, Figure 6 and Figure 7 are shown in Table 5, Table 6 and Table 7.

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
