# Peer review of "The Effect of Carbon Nanotubes on the Strength of Sand Seeped by Colloidal Silica in Triaxial Testing"

_materials, 2021, doi:10.3390/ma14206119_

Round 1

Reviewer 1 Report

materials-1377655-peer-review-v1

Title: The effect of carbon nanotubes on the strength of sand seeped by colloidal silica in triaxial testing.

The paper deals with the influence of carbon nanotube content on the shear strength of sands seeped with different concentration levels of colloidal silica, and proposes a formula for calculating the maximum shear strength.

The manuscript is a well-organised paper that introduce the subject matter, present the content in a clear and logical order, and concludes with a paragraph that reminds the reader of the main results that have been presented.

I think the topic of the paper could be interesting for international readers, and I suggest publishing it in the present form.

Author Response

No response is needed.

Reviewer 2 Report

The manuscript proposes a novel method for mixing carbon nanotubes and colloidal silica to seep through sand for sandy ground improvement. To do this authors set an experimental study to perform triaxial tests on several reinforced specimens considering different concentrations of colloidal silica and carbon nanotubes. The research experimental need further description and technical background which has not been given adequetly. In addition the state of the art and the research gap have not been given in a comprehensive manner.  Research elaborates on increasing the colloidal silica concentration while the shear strength significantly increases with the same carbon nanotube content. However, validation of the results are not discussed well. Please discussed on the results. How this can be improved? The formula to predict the concentration threshold of colloidal silica and shear strength rise and drop through increasing carbon nanotube content must be accompanied with further elaboration on its origins, background and citations. Further minor comments: 

Although the paper has appropriate length and informative content, several parts must be improved and written in better grammar and syntax. It would be essential if authors would consider revising the organization and composition of the manuscript, in terms of the definition/justification of the objectives, description of the method, the accomplishment of the objective, and results.

The paper is generally difficult to follow. Paragraphs and sentences are not well connected. Furthermore, I advise considering using standard keywords to better present the research. Improve the keywords to suite the methods, instead use the standard keywords not more than 8.

Please revise the abstract according to the journal guideline. It must be under 200 words. The research question, method, and the results must be briefly communicated. The abstract must be more informative.

I suggest having four paragraphs in the introduction for; describing the concept, research gap, contribution, and the organization of the paper. The motivation has the potential to be more elaborated. You may add materials on why doing this research is essential, and what this article would add to the current knowledge, etc.

The originality of the paper is not discussed well. The research question must be clearly given in the introduction, in addition to some words on the testable hypothesis.

Please elaborate on the importance of this work. Please discuss if the paper suitable for broad international interest and applications or better suited for the local application? Elaborate and discuss this in the introduction.

State of the art needs improvement. A detailed description of the cited references is essential. Several recently published papers are not included in the review section.

In fact, the acknowledgment of the past related work by others, in the reference list, is not sufficient. Consequently, the contribution of the paper is not clear. Furthermore, consider elaborating on the suitability of the paper and relevance to the journal. Kindly note that references cited must be up to date.    

Elaborate on the method used and why used this method.

Limitations and validation are not discussed adequately. The research question and hypothesis must be answered and discussed clearly in the discussion and conclusions. Please communicate the future research. The lessons learned must be further elaborated in the conclusion by discussing the results to the community and the future impacts. What is your perspective on future research?    

Reviewer 3 Report

W. Jin et al. describe the strength induced by CNTs reinforced in coloidal silica seeped into sand, for the purpose of sandy ground improvement. The research plan is generally good, and it follows closely that presented in ref. [42], by F. Schnaid et al. 

There are a number of small issues that I believe need to be addressed:

  • page 1, " The existed researches also .." needs to be reformulated;
  • page 4: Thy were the molds wrapped in moisture proof membranes and then immersed in water for 3 day curing? It seems counterintuitive.
  • same page (4), it seems odd that samples (c) and (d) look so different from (a) and (b), can the authors explain (Fig. 4)?
  • Formula (2) was deduced indeed in ref [42] based on a number of points' modelling, but formula (4) is not obvious and should be somewhere deduced (could also be placed in an SI), and formula (6) seems a bit stretched: analysis of Fig. 8 (which by the way has some wording missing from its title) doesn't necessarily show the linear trend, and i'm sure some polynomial would better fit that data; the projected point passed trough none of the experimental points...
  • As such, I'd advise revision of proposed formula (7), perhaps a few more test points would clarify the trend (Fig. 8)
  • It is also apparent that the prediction is too much off in case of sample seeped with 40 wt% colloidal silica with deviatoric stress at failure of 110 kPa (most points fall behind prediction, by quite some margin), which can indicate a different dependence
  • " the carbon nanotubes are entangled into ropes to bridge the crack, thereby suppressing crack propagation and increasing the macro strength of the specimen. " on page 11 - how do the authors make sure that the cracks are bounded by the CNTs? To this reviewer Fig 12 cannot be a proof of this claim.
  • " Besides, the formula of shear
    strength that varies with carbon nanotube content can be applied to site reinforcement design
    " - again, the mathematical model needs to be carefully checked, as suggested above.
